# Visiting Policies of Hospice Wards during the COVID-19 Pandemic: An Environmental Scan in Taiwan

**DOI:** 10.3390/ijerph17082857

**Published:** 2020-04-21

**Authors:** Ya-Chuan Hsu, Ya-An Liu, Ming-Hwai Lin, Hsiao-Wen Lee, Tzeng-Ji Chen, Li-Fang Chou, Shinn-Jang Hwang

**Affiliations:** 1Department of Family Medicine, Taipei Veterans General Hospital, No. 201, Sec. 2, Shi-Pai Road, Taipei 112, Taiwan; ych97160@gmail.com (Y.-C.H.); a3786923@gmail.com (Y.-A.L.); minghwai@gmail.com (M.-H.L.); a0921339856@gmail.com (H.-W.L.); sjhwang@vghtpe.gov.tw (S.-J.H.); 2School of Medicine, National Yang-Ming University, No. 155, Sec. 2, Linong Street, Taipei 112, Taiwan; 3Big Data Center, Department of Medical Research, Taipei Veterans General Hospital, No. 201, Sec. 2, Shi-Pai Road, Taipei 112, Taiwan; 4Department of Public Finance, National Chengchi University, Taipei 116, Taiwan; lifang@nccu.edu.tw

**Keywords:** COVID-19, health care surveys, hospices, severe acute respiratory syndrome coronavirus 2, visitors to patients

## Abstract

During an epidemic, almost all healthcare facilities restrict the visiting of patients to prevent disease transmission. For hospices with terminally ill patients, the trade-off between compassion and infection control becomes a difficult decision. This study aimed to survey the changes in visiting policy for all 76 hospice wards in Taiwan during the COVID-19 pandemic in March 2020. The altered visiting policies were assessed by the number of visitors per patient allowed at one time, the daily number of visiting slots, the number of hours open daily, and requisites for hospice ward entry. The differences in visiting policies between hospice wards and ordinary wards were also investigated. Data were collected by reviewing the official website of each hospital and were supplemented by phone calls in cases where no information was posted on the website. One quarter (n = 20) of hospice wards had different visiting policies to those of ordinary wards in the same hospital. Only one hospice ward operated an open policy, and in contrast, nine (11.8%) stopped visits entirely. Among the 67 hospice wards that allowed visiting, at most, two visitors at one time per patient were allowed in 46 (68.6%), one visiting time daily was allowed in 32 (47.8%), one hour of visiting per day was allowed in 29 (43.3%), and checking of identity and travel history was carried out in 12 wards (17.9%). During the COVID-19 pandemic, nearly all hospice wards in Taiwan changed their visiting policies, but the degree of restriction varied. Further studies could measure the impacts of visiting policy changes on patients and healthcare professionals.

## 1. Introduction

Given its geographical proximity and close economic and cultural ties with China, Taiwan has faced an enormous epidemic threat and has been on high alert since the emergence of severe acute respiratory syndrome coronavirus 2 (SARS-CoV-2) and the associated coronavirus disease (COVID-19) in Wuhan, Hubei, China at the end of December 2019. Taiwanese authorities have taken wide-ranging early precautions to contain the outbreak. One of the many preventive efforts was an alteration of visiting policies at almost all levels of healthcare facilities, following guidance issued by the Taiwan Centers for Disease Control (Taiwan CDC) [1]. Policies for patient visits have been discussed in previous studies; however, these have mostly focused on adult or pediatric intensive care units. Limited studies have been conducted in the setting of hospice wards, where end-of-life care involves the company of family members or loved ones. Studies have highlighted that visiting a patient is a positive and effective way to increase a sense of well-being in the patient and to help families cope with stress, loss, and further bereavement [2,3]. Therefore, based on cultural factors and evidence-based benefits, hospice wards in Taiwan tend to operate an open and liberal visiting policy, even allowing access to pets [4]. However, studies have also shed light on the negative effects of unrestricted visiting of patients, such as exacerbating patients’ physical, emotional, and spiritual distress, disrupting medical care delivery to patients, interrupting daily ward routines, and increasing the possibility of disease transmission between patients and visitors [5]. Thus, a dilemma arises for healthcare providers and policymakers during an epidemic. This study aimed to investigate changes to visiting policies in all hospice wards in Taiwan during the COVID-19 pandemic. We wanted to illustrate how hospice wards made the trade-off between compassion and infection control in the face of an impending crisis. The results have the potential to inform best practice for hospice visiting policies in the future.

## 2. Materials and Methods

### 2.1. Data Collection

This environmental scan was performed for the period 16–25 March 2020. The first case of COVID-19 in Taiwan was announced on 21 January. As of 25 March, there were 235 confirmed cases. There were 81 hospice wards as of February 2020, according to the directory edited by the Taiwan Academy of Hospice Palliative Medicine (TAHAM). The website provided information on all 81 hospice wards, including the name of the hospital, the number of beds, and a link to a webpage for each hospital’s dynamic bed capacity status. We then searched the official website of each hospital to check the current visiting policy for the hospice. We found that almost all of the hospitals displayed a revised new visiting policy for ordinary or intensive care units on their websites. If the changes in visiting policy were not mentioned on the website, we then made a telephone call to the ward for such information. Telephone numbers were obtained through the official websites of the hospitals. We confirmed the altered visiting policy with the head nurse or medical director of each hospice ward. The altered visiting policies were assessed based on the number of visitors per patient allowed at one time, the daily number of visiting slots, the number of hours open daily, and requisites for hospice ward entry.

### 2.2. Study Design and Data Extraction

A simple database (Microsoft Excel) was constructed to store the data on visiting policies at the included 76 hospice wards as well as ordinary wards at the same hospitals. The characteristics of the hospice wards were categorized according to the type of hospital (public or private) and the ward size (less than seven beds was classified as a small ward, 7–10 beds as a medium ward, and more than 10 beds as a large ward). We also evaluated the differences between the visiting policies of hospice wards and ordinary wards. We classified the restriction on the number of visitors per patient at one time into three categories (one, two, and unlimited visitors) and the daily number of visiting slots into three groups (one, two, and unlimited visiting slots). The total hours open per day was calculated by adding up the duration of each visiting slot and stratifying the total into six categories (less than 1 h, 1 h, 1.5 h, 2 h, more than 2 h, and unlimited hours). The requisites for entry to the ward were typically applied for all visitors at the main entrance of the hospital, including checking recent travel, occupation, contact and cluster information (TOCC history), monitoring body temperature by infrared thermal devices, and using hand sanitizer for hand hygiene. We confirmed whether any identity documents, such as identification cards (ID cards), National Health Insurance cards (NHI cards), or Taiwan resident certificates, were required to additionally check identity or travel history before visiting the hospice.

Based on prior studies, we defined the visiting policy as one of three types according to the degree of restriction: an open policy sets no restrictions at all, a structured visiting policy imposes a flexible approach, and a restricted policy means that patients on the ward are not allowed visitors [6,7].

### 2.3. Statistical Analysis

Descriptive statistics were calculated. Because our study included all hospice wards, we did not use inferential statistics for sampling. Only counts and percentages are displayed.

## 3. Results

### 3.1. Characteristics of Hospice Wards and Their Visiting Policies

Of 81 hospice wards in Taiwan, five hospice wards were excluded from our study due to being shared by two hospitals, the ward being temporarily shut down for infection control, being an unopened ward, or being a ward without current inpatients.

Of the 76 wards under analysis, 35 wards (46.1%) were in public hospitals and 41 wards (53.9%) were in private hospitals. Ward size was defined by the number of beds. Of the included wards, 17 wards (22.3%) contained less than 7 beds, 24 wards (31.6%) had 7–10 beds, and 35 wards (46.1%) had more than 10 beds.

During the COVID-19 pandemic, nearly all hospice wards (98.6%) in Taiwan changed their visiting policies to varying degrees. Only one hospice ward maintained an unchanged visiting policy and was open at all hours. The majority (86.8%) of hospice wards used a structured visiting policy. Visiting was completely stopped in nine hospice wards (11.8%) (Table 1).

### 3.2. Restrictions on the Visiting of Patients in Hospice Wards Vary with Hospice Ward Characteristics

The details of altered visiting policies at 67 hospice wards that allowed visitors, according to the type of hospital and ward size, are presented in Table 2. Over two-thirds (n = 46) of hospice wards permitted, at most, two visitors per patient at one time, and 25.4% (n = 17) allowed one visitor. Only four hospice wards (6%) had no limit on the number of visitors. Restrictions on the number of visitors per patient at one time were similar across all hospital types and ward sizes. 

The number of daily visiting slots allowed was predominantly one (32 wards; 47.8%) or two (28 wards; 41.8%) per day. Only seven wards (10.4%) set no limit on the number of daily visiting slots. For hospice wards in private hospitals or small wards, two visiting slots was the most common policy (18/35, 51.4% and 8/17, 47.1%, respectively). However, hospice wards in public hospitals or large wards tended to have only one daily visiting slot.

Visiting policies that allowed one hour of total daily visiting time were the most common (29 wards; 43.3%). However, hospice wards in private hospitals most commonly allowed two hours of visiting daily (16/35, 45.7%).

To gain entry to hospice wards, identification documents (ID card, NHI card, or Taiwan resident certificates) were required to be presented at 12 wards (17.9%) to check visitor identity and assess travel history.

### 3.3. Visiting Policies Differ between Hospice Wards and Ordinary Wards

Out of the 76 hospice wards analyzed in this study, 26.3% (n = 20) applied a different visiting policy to the ordinary wards of the same hospital during the COVID-19 pandemic. The differences between these two types of wards could be categorized into four types. Fifteen hospitals (75%) allowed visiting at the hospice ward but stopped visiting in ordinary wards. Only one hospital (5%) stopped patient visits in the hospice ward but permitted visiting in ordinary wards. Visiting of patients was allowed at both types of wards in the remaining four hospitals, of which three (15%) differed between the hospice and ordinary wards in the allowed numbers of visitors at one time and one (5%) differed in the total number of hours open per day (Table 3).

## 4. Discussion

### 4.1. Principal Findings

This study provides a comprehensive picture of the visiting policies of hospice wards in Taiwan during the COVID-19 pandemic. We showed that nearly all hospice wards implemented changes to their normal liberal visiting policies in response to the threat presented by the COVID-19 pandemic. As of March 2020, most of the hospice wards had imposed a structured visiting policy to strike a balance between compassion and infection control. Although approximately 10% of hospice wards used a restricted policy as the underlying principle of inpatient visiting, flexibility was offered in these wards in situations such as when a patient was in an unstable condition or dying, or when there was a need to conduct a family meeting. Among hospice wards that allowed visiting, most allowed a maximum of two visitors per patient at one time and one or two visiting slots per day. In general, hospice wards in private hospitals or small wards tended to have more liberal visiting policies with less restriction on the number of visitors per patient at one time, the daily number of visiting slots, and the total daily open hours. Identity documents are currently required for visiting hospice patients, to allow identity checks or travel history assessments.

### 4.2. Features of Visiting Policies at Hospice Wards

Visiting the patient is one of the most frequently identified needs of relatives caring for a loved one. Several studies have pointed out a significant correlation between the level of satisfaction with social support from relatives of a patient with a life-limiting disease and the degree of meaningfulness of the patient’s life [8,9,10,11,12]. Visiting of patients can also reduce levels of anxiety in patients and their families and foster communication among staff, patients, and families [3,13]. Therefore, with a growing recognition of the vital role that families play in end-of-life care, which has been characterized by “patient- and family-centered care”, hospice wards in Taiwan have previously advocated for an open or more liberal visiting policy with a more home-like environment than other types of wards [14]. However, we found that, as of March 2020, during the COVID-19 pandemic, there was almost a complete lack of open visiting policies at hospice wards, with nine wards (11.8%) stopping inpatient visiting completely. In other words, almost all (98.6%) of 76 surveyed hospice wards followed the guidance of the Taiwan CDC, which recommended that healthcare facilities restrict the duration of each visiting slot to a maximum of one hour with, at most, two visitors per patient at one time and avoid inpatient visiting if not necessary. Other nations implemented similar regulations during this period [15,16]. Studies outside the pandemic setting have not found significant infection of patients by visitors [17], but concern regarding transmission between patients and visitors persists in Taiwan and internationally.

The epidemiologic characteristics of COVID-19 are not yet fully understood. Current evidence shows some similarities between severe acute respiratory syndrome (SARS) and COVID-19, including person-to-person transmission through the respiratory droplet, contact, and probably fomite routes [18,19]. Based on tragic lessons learned from the SARS outbreak in 2003 and current knowledge about COVID-19, Taiwanese authorities have instigated early precautions to interrupt person-to-person transmission. Studies indicate that social distancing is particularly useful for delaying and broadening the peak of an epidemic until a vaccine is available, thereby reducing pressure on health services [20]. Visitors are also viewed as potential vectors in pathogen transmission in both healthcare facilities and communities during outbreaks of respiratory pathogens [21,22]. Theoretically, a restricted visiting policy might be an effective way to protect vulnerable populations, such as the elderly and those with underlying chronic or life-limiting medical conditions, from poor outcomes of COVID-19. However, our findings indicate that, though nine hospice wards (11.8%) adopted a restricted visiting policy, all of them emphasized flexibility and adaptability within such policies. Another finding of our study is that, of the hospice wards using a different visiting policy to ordinary wards in the same hospital, the majority (75%) allowed visiting in hospice wards but prevented visiting in ordinary ones. We could conjecture that the hospices perceive that they need a more human-based and customized visiting policy. Nonetheless, setting a suitable visiting policy for hospice wards remains a difficult decision-making process for healthcare personnel and policymakers. Studies to evaluate the effectiveness of differing visiting policies in protecting vulnerable patients during the COVID-19 pandemic could be valuable in determining appropriate policies in future scenarios.

### 4.3. Requisite for Hospice Ward Entry

In response to the threat of COVID-19, Taiwanese health authorities integrated each citizen’s travel history into the National Health Insurance Administration (NHIA) database from 13 January 2020. This feature allows personnel in healthcare systems to access users’ travel history records for the past 14 days by entering their National ID or inserting their NHI smart card into the system [23]. Because of this measure, ID documents have been required by more and more hospitals or wards for checking the eligibility of patients and visitors. We found that the presentation of ID documents was needed in only twelve (17.9%) out of the 67 hospice wards. This could be explained by the expectation of many hospice wards that identity or travel history checks would be done at the main entrances of hospitals. To decrease administrative tasks and most effectively use ward personnel, wards would therefore not duplicate identity document checks.

### 4.4. Limitations

There are several limitations to this study. First, visiting policies might change at any time with the progression of an infectious disease. We could only take a snapshot of visiting policies at hospice wards at a particular time (March 2020). Therefore, the results in this study may not correlate with those of another similar study at a different time. Second, in consideration of the characteristics of hospice patients and their families, there is more flexibility and adaptability within restricted visiting policies. Thus, we might be overestimating the number of restricted visiting policies and underestimating the number of structured visiting policies in practice. Third, we only used the publicly available information of hospice wards on the website of TAHAM to evaluate the associations between features of the altered visiting policy and characteristics of the hospice wards (the type of hospital and ward size). However, other variables such as the geographical area and how long the ward had been in operation might independently be associated with the degree of policy restriction. Finally, we only gave an overview of current visiting policies and did not explore the perceptions of patients, visitors, and staff members towards changes to visiting policies, which would be of value to provide a more comprehensive understanding of the impact of visiting policy changes on patients, visitors, and the multidisciplinary team. Further investigations are thus required to benefit the patients, visitors, and healthcare providers.

## 5. Conclusions

Nearly all hospice wards in Taiwan altered their visiting policies during the COVID-19 pandemic, but the degree of restriction varied. It is challenging to modify a visiting policy to strike a balance between concerns of patients and families on the one hand and the healthcare staff on the other. The impacts of current visiting policy changes on all members of the healthcare team, as well as patients and their families, should be further investigated in the future to evaluate the effectiveness of varying policies.

## Figures and Tables

**Table 1 ijerph-17-02857-t001:** Characteristics of 76 hospice wards and their visiting policy types.

Features	n	%
Types of visiting policy		
Restricted	9	11.8
Structured	66	86.8
Open	1	1.3
Types of hospital		
Public	35	46.1
Private	41	53.9
Number of beds		
<7 (Small)	17	22.3
7–10 (Medium)	24	31.6
>10 (Large)	35	46.1

**Table 2 ijerph-17-02857-t002:** Altered visiting policies at 67 hospice wards allowing visitors.

	Type of Hospital		Size of Ward			Total n = 67 (%)
	Public n = 32 (%)	Private n = 35 (%)	Small n = 17 (%)	Medium n = 21 (%)	Large n = 29 (%)	
Limit on the number of visitors per patient at one time						
One visitor	8 (25)	9 (25.7)	4 (23.5)	5 (23.8)	8 (27.6)	17 (25.4)
Two visitors	23 (71.9)	23 (65.7)	12 (70.6)	15 (71.4)	19 (65.5)	46 (68.6)
Unlimited	1 (3.1)	3 (8.6)	1 (5.9)	1 (4.8)	2 (6.9)	4 (6)
Limit on the daily number of visiting slots						
One visiting slot	17 (53.1)	15 (42.9)	5 (29.4)	10 (47.6)	17 (58.6)	32 (47.8)
Two visiting slots	10 (31.3)	18 (51.4)	8 (47.1)	10 (47.6)	10 (34.5)	28 (41.8)
Unlimited	5 (15.6)	2 (5.7)	4 (23.5)	1 (4.8)	2 (6.9)	7 (10.4)
Total daily open hours						
Less than 1 h	1 (3.1)	2 (5.7)	1 (5.9)	0 (0)	2 (6.9)	3 (4.5)
1 h	16 (50)	13 (37.1)	5 (29.4)	10 (47.6)	14 (48.3)	29 (43.3)
1.5 h	0 (0)	1 (2.9)	1 (5.9)	0 (0)	0 (0)	1 (1.5)
2 h	7 (21.9)	16 (45.7)	4 (23.5)	8 (38.1)	11 (37.9)	23 (34.3)
More than 2 h	3 (9.4)	1 (2.9)	2 (11.8)	2 (9.5)	0 (0)	4 (6.0)
Unlimited	5 (15.6)	2 (5.7)	4 (23.5)	1 (4.8)	2 (6.9)	7 (10.4)
Identity or TOCC check	4 (12.5)	8(22.9)	0 (0)	4 (19)	8 (27.6)	12 (17.9)

**Table 3 ijerph-17-02857-t003:** Differences in visiting policy between hospice and ordinary wards at 20 hospitals.

	n	%
Visiting banned in hospice wards but allowed in ordinary wards	1	5
Visiting banned in ordinary wards but allowed in hospice wards	15	75
Visiting allowed in both, but ordinary and hospice wards differ in:		
Number of visitors per patient at one time	3	15
Duration of visiting time	1	5

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
