# Peer review of "Visiting Policies of Hospice Wards during the COVID-19 Pandemic: An Environmental Scan in Taiwan"

_ijerph, 2020, doi:10.3390/ijerph17082857_

Round 1

Reviewer 1 Report

Abstract

It`s necessary to describe briefly the main methods 

Introduction

You must end with the main aim, not with [54]The results have the potential to inform best practice for hospice visiting policies in the future.

You must explain the characteristics of Hospice Wards

Materials and Methods

It`s very important explain the impact of COVID-19 in this date. Has contagion curve passed? You should remove the URL [59,60]

You should explain the Statistical Analysis, it is not enough.

Results

How many hospice wards are in Taiwan?

When you say: [105] The majority (66/76, 86.8%)... You should put only 86.8%.

Reviewer 2 Report

This manuscript is of interest to health care providers, healthcare organizations and countries around the world. The greatest challenge of this manuscript was that is was identified as being a survey when it appears to have been an environmental scan. However, if this is the case, the methods used are not clearly laid out. 

A survey usually has specific questions that are asked of each organization and is not undertaken without receiving Ethical Approval from the appropriate body/organization. The authors indicated in the Limitations that only publicly available information of hospice wards on the website of TAHAM were used to evaluate the association between features of the altered visiting policy and characteristics of the hospice wards (the type of hospital and ward size). This may have been true; however, the authors indicated in the Data Collection that very few details of the hospice ward policies were mentioned on the website so they conducted a comprehensive telephone survey across all hospice wards in Taiwan.

In looking at Table 2. entitled "Altered visiting policies at 67 hospice wards allowing visitors", please note that the Identity or TOCC check is incorrect in columns that read the Size of the Ward - Medium and Large as the number indicated only adds up to 11 and not 12.

The following references were not cited in any aspect of the manuscript: References 2-5, 9 and 26.

I would encourage the authors to reflect upon these comments. As a reviewer, I am not sure how the issue of Ethical Approval can be resolved after the data has been collected. That being said, this information is of interest to health care providers, healthcare organizations and countries around the world as they work to support those in hospice care and their families during this world-wide pandemic.      

Round 2

Reviewer 1 Report

All comments have been resolved. Congratulations on your article

Author Response

We sincerely thanks for the reviewer's precious comments and recommendations on our manuscript.

Reviewer 2 Report

The authors have done a good job of addressing the noted concerns in the previous version of the manuscript. However, there remain a couple of things that need to be corrected before consideration is given for publication. These are:

  1. Under 2.1 Data Collection - This descriptive study should read "This environmental scan....".
  2. Reference #3 is incomplete as the name of journal does not appear to be listed nor the year of publication.  
